# The feasibility of a visuo-cognitive training intervention using a mobile application and exercise with stroboscopic glasses in Parkinson's: Findings from a pilot randomised controlled trial

**Julia Das**[1,2], **Gill Barry**[1], **Richard Walker**[2], **Rodrigo Vitorio**[1], **Yunus Celik**[3], **Claire McDonald**[4], **Bryony Storey**[4], **Paul Oman**[5], **Rosie Morris**[1,2], **Samuel Stuart**[1,2,6]*

**1** Department of Sport, Exercise & Rehabilitation, Northumbria University, Newcastle upon Tyne, United Kingdom, **2** Northumbria Healthcare NHS Foundation Trust, North Tyneside General Hospital, North Shields, United Kingdom, **3** Department of Computer and Information Sciences, Northumbria University, Newcastle upon Tyne, United Kingdom, **4** Gateshead Health NHS Foundation Trust, Gateshead, United Kingdom, **5** Department of Mathematics, Physics & Electrical Engineering, Northumbria University, Newcastle upon Tyne, United Kingdom, **6** Department of Neurology, Oregon Health & Science University, Portland, Oregon, United States of America

* samuel2.stuart@northumbria.ac.uk

## Abstract

### Background

There is currently no pharmacological treatment for visuo-cognitive impairments in Parkinson's disease. Alternative strategies are needed to address these non-motor symptoms given their impact on quality of life. Novel technologies have potential to deliver multimodal rehabilitation of visuo-cognitive dysfunction, but more research is required to determine their feasibility in Parkinson's.

### Objective

To determine the feasibility and preliminary efficacy of a home-based, technological visuo-cognitive training (TVT) intervention using a mobile application and exercise with stroboscopic glasses compared to non-technological care in people with Parkinson's.

### Methods

This 18-month, parallel, two-arm pilot trial took place between July 2021-December 2022. Participants were community-dwelling individuals with a diagnosis of Parkinson's, aged over 50 years. Participants were randomly allocated to one of two active four-week interventions, TVT (n = 20) or standard care (SC) (n = 20). A physiotherapist delivered 8 home visits over 4 weeks, lasting 45–60 mins. Participants were evaluated at baseline and then on completion of the intervention. Primary outcomes were feasibility of the study design and intervention (recruitment/retention, adherence, assessment time scale, equipment and safety).

**Data Availability Statement:** This study is registered with ISRCTN and the data presented in this study are available in the manuscript itself.

**Funding:** This work was funded by a Northumbria University PhD studentship in collaboration with Senaptec Ltd. (Beaverton, Oregon, USA) to JD. Additional support was also provided by the British Geriatric Society to JD, a Parkinson's Foundation post-doctoral fellowship for basic scientists (PF-PDF-1898) to SS and a clinical research award (PF-CRA-2073) to SS. The funders had no role in study design, data collection and analysis, decision to publish, or preparation of the manuscript.

**Competing interests:** Julia Das is funded by a Northumbria University PhD studentship in collaboration with Senaptec Ltd (Beaverton, Oregon, USA). The other authors have no relevant affiliations or financial involvement with any organization or entity with a financial interest in or financial conflict with the subject matter or materials discussed in the manuscript.

Exploratory outcomes included assessments of cognitive, visual, clinical and motor function. (Blinding of participants was not possible due to the nature of the intervention)

## Results

The recruitment rate was 60% (40/67), and the retention rate was 98% (39/40). Adherence to both arms of the intervention was high, with participants attending 98% of visits in the TVT group and 96% of visits in the SC group. 35% (9/20) of participants in the TVT group experienced mild symptoms associated with use of the stroboscopic glasses which included dizziness, queasiness and unsteadiness. There were minimal between group differences, with both interventions having positive effects on a variety of clinical, cognitive, and physical performance outcomes.

## Conclusions

Our findings suggest that home-based TVT with a physiotherapist is feasible in people with Parkinson's and could provide an alternative approach to addressing cognitive and motor dysfunction in this population. We make recommendations for future trials and invite ensuing studies to improve upon the design and utilise stroboscopic visual training and digital tools to investigate this emerging area of multimodal rehabilitation.

This trial was prospectively registered at ISRCTN (registration number: ISRCTN46164906; https://doi.org/10.1186/ISRCTN46164906).

### Author summary

Visual and cognitive dysfunction are common in Parkinson's disease (PD) and relate to increased falls risk and reduced quality of life. Vision and cognition are interrelated (termed visuo-cognition) and there are currently no effective medications to treat these symptoms. Non-pharmacological interventions for visuo-cognitive deficits are possible with modern, digital technology, but evidence for their effectiveness in PD is lacking. We aimed to find out whether visuo-cognitive training, using stroboscopic glasses and a mobile application, is feasible for people with PD. We recruited 40 people with PD, and randomly assigned them to one of two visuo-cognitive training groups: Technological visuo-cognitive training–TVT (n = 20), and standard, non-technological care–SC (n = 20). After baseline assessment, participants received visuo-cognitive training at home, from a physiotherapist, twice a week for 4 weeks. They were then reassessed on completion of the intervention. Adherence to both the interventions was high, and there was only one withdrawal from the study. Preliminary results showed that participants in both groups improved in a variety of clinical, cognitive, and physical performance outcomes. Our findings suggest that home-based TVT with a physiotherapist is feasible in people with Parkinson's and could provide an alternative approach to addressing cognitive and motor dysfunction in this population.

## Introduction

Parkinson's disease (PD) is a neurodegenerative condition characterised by motor symptoms (i.e., rigidity, bradykinesia, tremor, postural instability, gait deficit) and non-motor symptoms

(i.e., visual function deficits, behavioural changes, cognitive impairment, fatigue etc.) that progress over time [1]. Motor and non-motor symptoms may progress with PD to the point that there is a loss of independence, increased caregiver burden and reduced quality of life (QoL) [2,3].

Cognitive impairment is one of the most recognised non-motor symptoms in PD as deficits occur early and are common features of the disease, including impaired executive function, visuo-spatial ability, working memory and attention [4]. While less studied, other impairments such as sensory visual deficits are also common in PD, with up to 75% of people with PD reporting at least one visual symptom [5]. Visual impairments range from retinal changes, causing reduced visual acuity and contrast sensitivity, to more complex visual processes such as depth or motion perception [6].These motor and non-motor symptoms can be difficult to treat, as they can be refractory to dopaminergic therapy [7], therefore rehabilitation protocols have been previously examined to provide non-pharmacological intervention for these symptoms.

The focus of previous rehabilitation studies and evidence generation has primarily been on motor symptoms of PD, with emerging evidence that cognitive training may also be useful in PD [8–11], but evidence is lacking for rehabilitation of visual symptoms. Visual impairment is typically treated with corrective lenses that can treat refractive errors [12] or visual manipulation through changing the visual environment (i.e., lighting, lines on the floor etc.) or technology (e.g., virtual or augmented reality). However, previous rehabilitation protocols and research have not recognised that visual and cognitive processes are highly integrated (termed visuo-cognition) and that these influence motor symptoms simultaneously [13]. For example, our previous work has shown that visuo-cognitive processes directly and indirectly influence motor task performance in PD [14–16].

Visuo-cognitive impairments have been associated with motor deficits in people with PD, specifically postural instability and gait impairment [5,17–20]. This is further complicated by the fact that deficits in the ability to use proprioceptive feedback in people with PD result in greater reliance on impaired visuo-cognitive function to carry out motor tasks [5,21–25]. The combination of visuo-cognitive deficits and increased dependence on faulty visuo-cognitive information has a significant impact on motor symptoms in people with PD, particularly gait and balance [24,26,27]. Specifically, visuo-cognitive impairments reduce the ability to compensate for underlying motor dysfunction in Parkinson's, which can lead to a decline in daily activities, increased risk of falls and reduced quality of life [28–30].

There is a lack of pharmacological treatment options for visuo-cognitive deficits [31]. Previous studies have shown some improvement in visuo-cognitive function in people with PD using clinical interventions, such as eye movement training and exercises that use paper-based or gamified activities (i.e., throwing and catching, hand eye co-ordination tasks etc.) [32–34]. However, the treatments described in these trials often involve participants having to attend multiple training sessions per week at a laboratory/clinic/hospital or performing repetitive activities, such as pencil push ups several times per day [34]. These studies have also been limited by small sample sizes, strict inclusion criteria, and the burdensome protocols that resulted in significant drop-out rates and low participant uptake, which all limit the generalisability to clinical practice.

As a result of these limitations, there is growing interest in the development of visuo-cognitive training programmes provided via digital technologies to offer an alternative approach to traditional rehabilitation interventions. These can be carried out using modern digital technologies, such as computer or mobile applications [35–42]. Technologies can offer potential advantages over paper-based tasks in that they can provide real-time feedback, allow modulation of task progression and automatic recording of session parameters [43]. In addition to computerised training applications, novel devices also have the potential to enhance traditional visuo-cognitive training approaches. For example, Shalmoni & Kalron (2020) demonstrated

that stroboscopic eyewear could enhance information processing speed immediately after training in people with multiple sclerosis [44].

As awareness of how visuo-cognitive processes directly and indirectly influence motor task performance in PD increases, interest in the potential of multimodal visuo-cognitive training strategies to simultaneously target visual, cognitive and motor abilities has grown [8–10,45]. To date, no studies have examined the combined use of stroboscopic visual training with a cognitive training application in PD (here termed technological visuo-cognitive training ("TVT")). Therefore, the aims of the present study were to; 1) Investigate the feasibility of home-based TVT compared to non-technological visuo-cognitive training (here termed standard care ("SC")) in PD; 2) Provide initial efficacy data of home-based TVT compared to SC in PD; and 3) To provide data to inform future larger scale clinical trials.

This study conforms to the recommendations for reporting the results of pilot feasibility studies which are adapted from the CONSORT Statement (S1 File) [46].

## Methods

### Ethical approval and registration

The design of this study conforms to the principles outlined in the Declaration of Helsinki. Ethics approval was obtained from the South Central-Berkshire B Research Ethics Committee (ref 21/SC/0042) and Northumbria University (Ref: 27828). Participation in the study was voluntary and required the written informed consent from each participant. This trial was listed on the ISRCTN registry with study ID ISRCTN46164906 on 21 April 2021.

### Design

The full protocol for this study has been previously published [47] and the trial protocol can be viewed in S2 File. Participants underwent baseline assessments in the Physiotherapy Innovation Lab at Northumbria University and were randomly assigned to either the TVT or the SC group. After completing four weeks of home-based interventions, participants were then reassessed in the Lab.

### Participants

40 people with PD were recruited to the study. Recruitment was through Movement Disorders Clinics based in the Northeast of England, the Parkinson's UK Research Support Network and the National Institute for Health Research–Central Portfolio Management System Dementias & Neurodegenerative Diseases Research Network (DeNDRoN). Eligibility criteria are included in the study protocol in S2 File.

### Intervention

The interventions are described in detail in the previously published protocol [47] and outlined in S3 File. In brief, a physiotherapist (JD) delivered 8 home visits over 4 weeks, with each visit lasting 45–60 minutes. The intervention sessions were planned at the participants' convenience and delivered when participants were in their "*on*" phase (within 60 minutes of dopaminergic medication intake). On the first home visit, the physiotherapist determined an appropriate location for the participant to undertake the training, taking into consideration potential trip hazards, adequate space, and lighting. The physiotherapist monitored participants' progress during each visit, factoring in rest periods as required.

The app-based training consisted of a series of adaptive drills which automatically increased in difficulty based on participant performance [48]. The SC arm was adapted from

interventions used in previous vision and cognitive therapy trials, as well as freely available online resources [49–51]. The SC interventions included non-technological strategies which were designed to mirror the TVT arm for content and timescale. Pen and paper and game-based activities were undertaken involving a variety of researcher-led visuo-motor, attention and perceptual tasks which were made progressively more challenging by adding time limits and increasing complexity of tasks [10,49,52]. Both groups undertook the same hand-eye co-ordination, balance, and gait exercises either with or without the stroboscopic glasses depending on group allocation. (A full description of the interventions according to the TIDieR framework is available in S4 File).

## Outcome measures

### Feasibility Outcomes

The primary outcome measures assessed feasibility of the study design and feasibility of the technological intervention. Data were collected in relation to the following indicators:

### Recruitment and retention

To determine whether the recruitment methods yielded the desired sample size (n = 40] in the time allotted (18 months), the number of people who were approached to take part was recorded, along with the number who consented and reasons for non-participation. Any drop-outs were logged and reported to refine the recruitment process for future larger trials [53]. The flow of participants in this study is shown in Fig 1.

### Adherence

Data on attendance and adherence to the interventions were collected by the researcher during the study period. Adherence rate was calculated as the ratio between the number of sessions completed and the total number of visits scheduled (8 sessions). Reasons for non-attendance and failure to complete a training session were also recorded.

### Assessment time scale

Feasibility of the assessment time scale was based on the number of participants whose follow-up data were collected within seven days of completing the intervention.

### Equipment

One pair of stroboscopic glasses was used throughout the study, with settings operated by the researcher either directly by buttons on the arm of the glasses or remotely via Bluetooth pairing with a mobile phone (Fig 2A). One tablet device (8th Generation iPadOS 8th Version 16) was used to deliver the app-based visuo-cognitive training drills (Fig 2B) [48]. The researcher kept a log of any issues that were encountered with the stroboscopic glasses and mobile app in relation to device function and connectivity. Participants were asked to rate the usability of the Senaptec application after initial use and then again at the end of the intervention period by completing the ten item System Usability Scale questionnaire (SUS) [54].

### Safety

The safety of the TVT intervention was assessed on the basis of adverse events (such as falls) or negative symptoms associated with the stroboscopic effect. At each home visit, the researcher logged any symptoms that were experienced during the session and questioned the

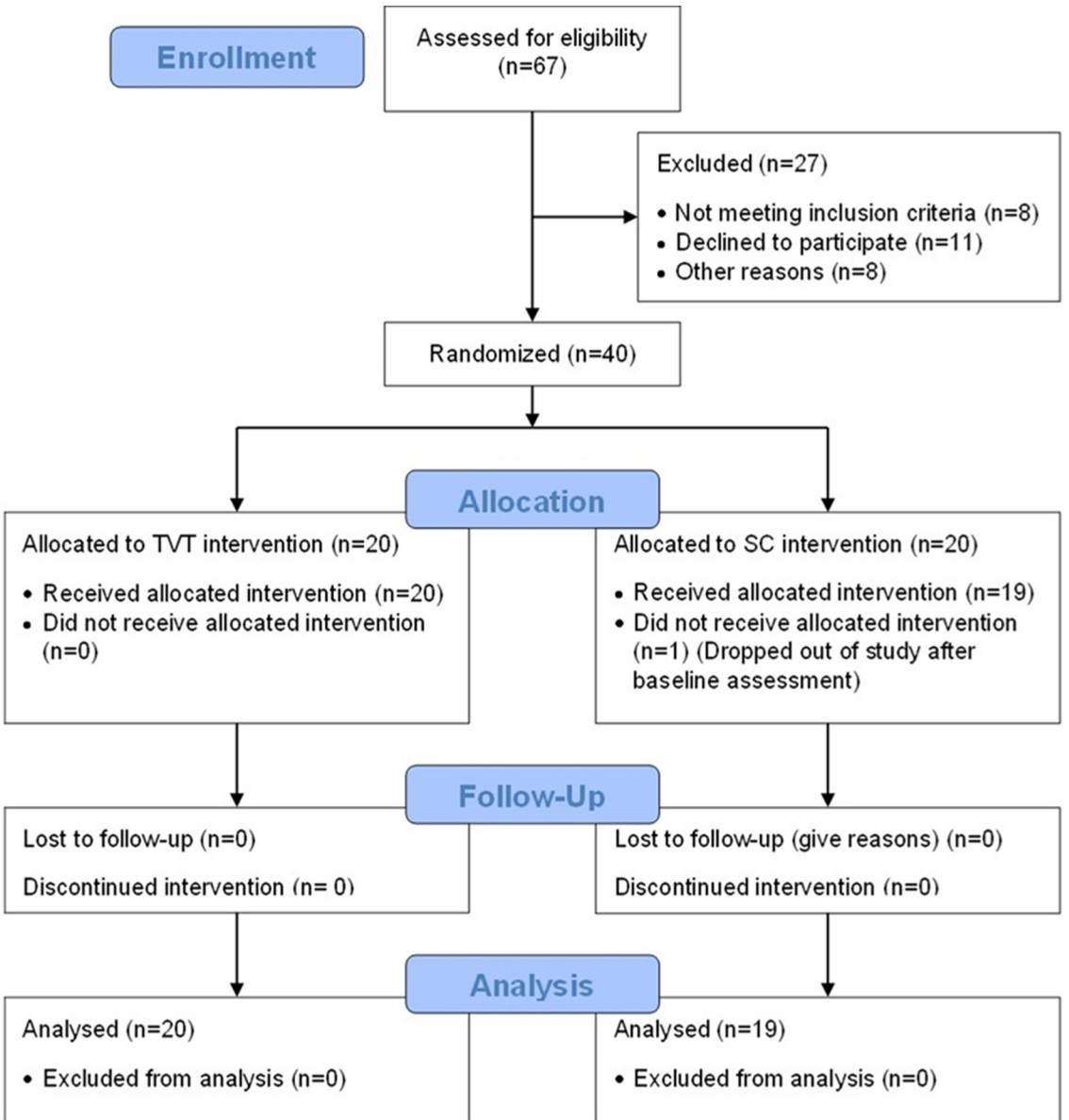

**Fig 1. CONSORT flow diagram illustrating recruitment, randomization, and tracking of participants over the course of the study.**

participants about the occurrence of any symptoms experienced between visits. Any symptom that was triggered by the intervention and resulted in the participant having to discontinue a particular activity during the training (or prevented participation in subsequent sessions) was recorded.

## Preliminary Efficacy Outcomes

Outcome measures were collected at baseline and post-intervention and can be viewed in the previously published study protocol paper [47]. In brief, the primary variable of interest was the Trail Making Test (TMT), due to the visuo-cognitive nature of the assessment (i.e., need for integrated visual and cognitive processing to complete the tasks) [47]. The TMT was

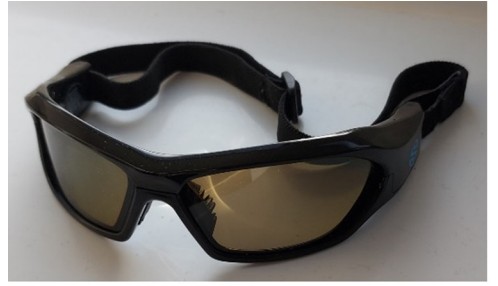
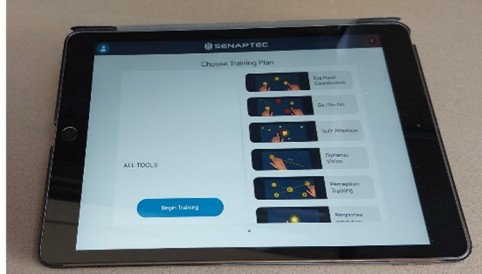

A Stroboscopic glasses                    B Senaptec application

**Fig 2. Stroboscopic glasses and tablet device used to deliver intervention.**

administered in two parts [55]. Part A of the test involved participants drawing lines to connect circles numbered from 1 to 25 in numerical order. In part B, participants were asked to connect an array of numbered and lettered circles in numerical and alphabetical order by alternating between numbers and letters. The TMT (B-A) score is calculated as the difference between TMT-B and TMT-A times and is considered a measure of cognitive flexibility [56]. Other outcomes were included to examine what cognitive, visual, motor or clinical outcomes may be useful for future studies.

*Cognitive outcomes*: Attention measured via a computerized button pressing test within Matlab involving simple (SRT) and choice reaction time (CRT) [57], and the Senaptec Sensory Station (SSS) reaction time task [48]. Visuo-spatial ability was measured with the Judgement of Line Orientation Test [58], SSS eye-hand co-ordination task and multiple object tracking. Working memory was measured with the forward digit span and SSS perception span tasks.

*Motor outcomes*: Motor symptoms were measured with the unified Parkinson's disease rating scale (MDS-UPDRS) section III [59]. Clinical balance analysis was performed with the Mini-BESTest [60] and timed up and go test [61]. Spatiotemporal gait (gait speed, foot strike angle, stride length, stride time, turn velocity) and balance (sway, jerk, velocity) characteristics were recorded via wearable inertial measurement units (Opal, APDM Wearable Technologies, USA) used during standing and walking. Gait and balance characteristics were extracted from the inertial sensors using the Mobility Lab software, V2 (APDM, USA) [62].

*Visual outcomes*: Visual acuity was measured with the LogMar visual acuity eye chart, SSS visual clarity and SSS near-far quickness. Contrast sensitivity was measured with the LogCS eye chart and SSS LogCS, CS-6 and CS-18 tasks [48].

*Self-reported clinical outcomes*: Fear of falling [63], fatigue severity [64] and QoL [65] were measured using self-report questionnaires.

## Statistical Analyses

Statistical analyses were conducted using SPSS Statistics for Windows, version 28 (IBM Corporation]. Due to the small sample size and the distribution for some variables not meeting assumptions of normality, non-parametric analyses were used. To compare baseline characteristics between each group (TVT and SC), a Mann Whitney U test was used for continuous variables and a chi-square test for categorical variables. Data are presented as median and interquartile range. To explore within group changes from baseline to final assessment, a Wilcoxon Signed Ranks test was performed. A change value for each parameter was calculated by subtracting the pre-intervention score from the post- intervention score. Change values for the two groups were compared with the Mann Whitney U test. Estimated effect sizes were

calculated using the formula ($Z/\sqrt{N}$). Magnitude of effect size was interpreted as small ($<0.30$), medium ($0.31$–$0.50$), and large ($>0.51$) as proposed by Cohen [66].

## Results

### Participants

Demographic participant background information by intervention group is presented in Table 1. Participants in the two groups were well-matched for demographic and clinical characteristics, motor, cognitive and visual functions (Table 1).

### Feasibility outcomes

**Recruitment and retention.** Recruitment to the study took place between July 2021 and October 2022. A total of 67 potential participants were identified and assessed for eligibility. Of these, 40 consented to participate, resulting in a 60% (40/67) recruitment rate. Of those participants who were approached but did not take part in the study, six did not respond to follow-up contact, nine did not meet eligibility requirements (i.e., history of migraines, n = 4; comorbidities limiting participation, n = 4; unpredictable "off" periods, n = 1), three declined due to other commitments (e.g. caring responsibilities, work), one was uncomfortable with the prospect of home visits and a further eight did not provide a reason. One participant (male) withdrew after baseline assessment due to issues unrelated to PD, resulting in a 98% (39/40) retention rate (Fig 1).

Recruitment predominantly took place from movement disorders clinics at Northumbria Healthcare NHS Foundation Trust (n = 24, 60%). Additional participants were recruited through the Parkinson's UK Research Support Network (n = 7, 18%), DeNDRoN (n = 4, 10%) and Gateshead Health NHS Foundation Trust (n = 5, 13%).

**Adherence.** The adherence rate was high in both arms of the study, with participants attending 98% of visits in the TVT group (mean 7.9±0.4) and 96% of visits in the SC group (mean 7.7±0.6). All participants took part in at least 6 out of the 8 visits and there was no difference in attendance between groups (p = 0.299). Reasons for the missed visits are presented in Table 2.

**Assessment time scale.** All follow-up assessments were conducted within seven days of completing the final intervention.

**Equipment.** Both the tablet and stroboscopic glasses operated with built-in batteries which lasted for a full day of home visits without requiring recharging. All participants in the TVT arm had home internet access which allowed the app to run, and no significant operational issues were encountered.

Participants rated the usability of the Senaptec application on the SUS [54]. After first use of the app, the mean score for the SUS (SUS; 0–100 range, higher scores indicating better performance) was 80.5±18 points (indicating "good" usability). After 4 weeks of use, the mean SUS was 88.6±11.3 points (indicating "excellent" usability)

**Safety.** No significant adverse events resulting in the need for medical attention or withdrawal from the study occurred. Four participants (two in each group) reported joint pain related to pre-existing conditions which affected their ability to perform exercises in standing. Five participants (four in the TVT group) experienced shortness of breath on exertion which limited their exercise tolerance. While none of these symptoms were severe enough to exclude these participants from the study, the physiotherapist reduced the intensity of the exercises and increased rest periods as necessary in these cases. One participant in the TVT group reported a fall that occurred due to a mechanical trip at home the day before a study visit, resulting in superficial bruising to the arms and head. The injuries did not require medical

**Table 1. Participant characteristics at baseline.**

| | | Total (n = 40) | TVT group (n = 20) | SC group (n = 20) | p |
|---|---|---|---|---|---|
| Demographic | Age, years (SD) | 70.5 (59.0–77.0) | 71.5 (62.5–77.0) | 69.0 (58.3–77.8) | 0.735 |
| | Gender, male/female | 30 (75)/10 (25) | 15 (75)/5 (25) | 15 (75)/5 (25) | 1.000 |
| | Living status | | | | 0.605 |
| | Lives alone | 4 (10) | 1 (5) | 3 (15) | |
| | Lives with spouse | 36 (90) | 19 (95) | 17 (85) | |
| | Education, years | 11.0 (10.25–14.75) | 11.5 (10.0–14.0) | 11.0 (11.0–16.5) | 0.751 |
| | Depression scale (GDS-15) | 3 (1, 4) | 2.5 (0.3–4.0) | 3.0 (1.3–6.3) | 0.257 |
| | Falls efficacy scale (FES-I) | 27.0 (20.3–35.5) | 26.0 (19.3–31.0) | 30.0 (21.0–44.8) | 0.110 |
| | Recurrent faller, yes/no[a] | 7 (18)/33 (83) | 4 (20)/16 (80) | 3 (15)/17 (85) | 0.687 |
| Clinical | H&Y stage | | | | 0.661 |
| | 1 | 9 (23) | 4 (20) | 5 (25) | |
| | 2 | 19 (48) | 10 (50) | 9 (45) | |
| | 3 | 12 (30) | 6 (30) | 6 (30) | |
| | PD duration, years | 4.5 (1.5–10.0) | 4.5 (3.0–10.0) | 4.5 (0.8–9.8) | 0.871 |
| | UPDRS Part III (motor) | 31.0 (20.0–47.5) | 39.0 (20.3–48.3) | 25.5 (19.3–46.8) | 0.525 |
| | FOGQ | 0 (0–13) | 0 (0–13) | 0 (0–14.5) | 0.988 |
| Cognition | MoCA | 27 (25–29) | 26 (25–28) | 28.0 (26.0–29.0) | 0.241 |
| | PDAQ-15 | 49 (41–55) | 50 (44.3–56.8) | 47 (36–53) | 0.136 |
| Executive function | CLOX 1 | 12.0 (11.0–13.0) | 12.0 (11.0–12.8) | 13.0 (12.0–13.8) | 0.093 |
| Visuo spatial ability | CLOX 2 | 14 (13–14) | 14 (12.3–14) | 14 (13–14) | 0.873 |
| | JLO | 20.0 (14.0–24.8) | 19.5 (13.3–25.0) | 20.0 (18.3–23.0) | 0.935 |
| Working memory | Seated forward digit span | 6.0 (5.0–7.0) | 6.0 (5.0–6.0) | 6.5 (5.3–7.0) | 0.072 |
| Visual function | Visual acuity[b] | 0.12 (0.04–0.23) | 0.14 (0.04–0.30) | 0.12 (0.0–0.20) | 0.204 |
| | Contrast sensitivity[c] | 1.46 (1.37–1.56) | 1.50 (1.33–1.56) | 1.44 (1.40–1.62) | 0.673 |

Continuous data are presented as median (IQR) and nominal data as proportions (percentages).

**Abbreviations**: GDS, Geriatric Depression Score; FES, Falls Efficacy Scale; H&Y, Hoehn and Yahr score; PD, Parkinson's disease; UPDRS, Unified Parkinson's Disease Rating Scale; FOGQ, Freezing of Gait Questionnaire; MoCA, Montreal Cognitive Assessment; PDAQ-15, Parkinson's Daily Activities Questionnaire; CLOX, Royall's Clock Drawing Test; JLO, Benton's Judgement Line of Orientation Test.

[a]Participants who had experienced ≥2 falls during the previous 12 months were classified as recurrent fallers

[b]Binocular visual acuity measured using LogMAR chart

[c]Binocular contrast sensitivity measured using Mars Chart

**Table 2. Adherence to intervention.**

| TVT group | | | Standard group | | |
|---|---|---|---|---|---|
| | Total visits (max. no. = 8) | Reason for missed visit(s) | | Total visits (max. no. = 8) | Reason for missed visit(s) |
| PD01 | 7 | Covid | PD04 | 8 | N/A* |
| PD02 | 7 | Fatigue/unwell | PD06 | 8 | N/A* |
| PD03 | 8 | N/A* | PD07 | 8 | N/A* |
| PD05 | 8 | N/A* | PD08 | 7 | Participant availability |
| PD11 | 8 | N/A* | PD09 | 7 | Participant availability |
| PD12 | 8[a] | N/A* | PD10 | 8 | N/A* |
| PD14 | 8 | N/A* | PD13 | 8 | N/A* |
| PD15 | 8 | N/A* | PD17 | 7 | Researcher availability |
| PD16 | 7 | Researcher availability | PD19 | 8 | N/A* |
| PD18 | 8[b] | N/A* | PD20 | 8 | N/A* |
| PD21 | 8 | N/A* | PD23 | 8 | N/A* |
| PD22 | 8 | N/A* | PD25 | N/A | Drop out |
| PD24 | 8 | N/A* | PD28 | 8 | N/A* |
| PD26 | 8 | N/A* | PD29 | 8 | N/A* |
| PD27 | 8[c] | N/A* | PD30 | 6 | Last minute holiday |
| PD31 | 8 | N/A* | PD34 | 7 | Unwell |
| PD32 | 8 | N/A* | PD35 | 8 | N/A* |
| PD33 | 8 | N/A* | PD37 | 8 | N/A* |
| PD36 | 8[d] | N/A* | PD38 | 8 | N/A* |
| PD40 | 8 | N/A* | PD39 | 8 | N/A* |
| Mean (SD) | 7.9 (0.4) | | Mean (SD) | 7.7 (0.6) | |

*N/A: Not applicable

[a]This participant exercised without the strobe glasses on visit 5 as a precautionary measure after experiencing prolonged queasiness. No further symptoms were observed.

[b]This participant did not exercise or wear the strobe glasses on the last two visits due to undergoing cataract surgery.

[c]This participant exercised without the strobe glasses on one visit due to high weather temperatures.

[d]This participant exercised without the strobe glasses as a precautionary measure on one visit following a minor head injury sustained the previous day as a result of a fall unrelated to the study.

attention but, as a precautionary measure, the participant performed exercises without the stroboscopic glasses on during the following visit.

35% of participants in the TVT group experienced mild symptoms associated with use of the stroboscopic glasses which included dizziness (n = 3), queasiness (n = 3) and unsteadiness (n = 1). In the absence of existing guidelines in clinical populations and in the interests of safety, the decision was made to remove the strobe glasses at the first indication of unwanted symptoms. In all but one case symptoms resolved on removal of the glasses and exercises could be resumed following a short rest. The exception to this was the participant who experienced mild queasiness after performing exercises involving 180 degree turns which did not resolve immediately on removal of the glasses. As a precautionary measure, the physiotherapist avoided rotational exercises during subsequent sessions and the participant was able to continue training with no further symptoms. No negative symptoms were experienced by participants in the SC group, exercising under normal visual conditions. No adverse effects were experienced as a result of using the mobile application, although the physiotherapist did provide postural feedback to ensure participants were well positioned during the training.

## Preliminary efficacy outcomes

Table 3 shows median scores for baseline and follow-up, within group and between group changes as a result of the intervention, for cognitive and visual outcomes. The primary efficacy outcome of TMT improved across both the TVT and SC groups. The TVT group showed medium effects for improvements in the TMT A (effect size: 0.38; p = 0.017) and TMT B (effect size: 0.40; p = 0.013). The SC showed medium effects for improvements in the TMT B (effect size: 0.41; p = 0.011) and the TMT (B-A) (effect size: 0.44; p = 0.006).

There was a trend towards improvements across a range of other cognitive measures in both groups. Working memory, measured by the Forward Digit Span (FDS), showed medium effect sizes for improvements in the TVT group (effect size: 0.31; p = 0.053) and in the SC group (effect size: 0.37; p = 0.021). The SC group showed a large effect size for improvements in computer-based CRT (effect size: 0.52; p = 0.002).

**Table 3. Baseline and follow-up scores (median and interquartile range), within group changes and between groups differences (p) for change over time for cognitive and visual outcomes.**

| Outcome | | TVT group | | | | | Standard group | | | | | Between group difference | |
|---|---|---|---|---|---|---|---|---|---|---|---|---|---|
| | | Pre | Post | Change score | p | Effect size | Pre | Post | Change score | p | Effect size | p | Effect size |
| **Cognitive outcomes** | | | | | | | | | | | | | |
| **Attention** | SRT* | 370.2 (329.2–434.6) | 332.6 (308.1–364.1) | -25.7 (-61.1–2.3) | 0.069 | 0.33 Medium | 351.6 (315.7–414.5) | 324.1 (303.6–397.0) | -9.3 (-43.1–11.5) | 0.094 | 0.28 | 0.718 | 0.06 |
| | SRT-CoV* | 0.2 (0.2–0.4) | 0.2 (0.1–0.2) | 0.0 (-0.2–0.0) | 0.172 | 0.25 | 0.2 (0.1–0.3) | 0.2 (0.2–0.2) | 0.0 (-0.0–0,0) | 0.943 | 0.01 | 0.224 | 0.21 |
| | CRT* | 530.0 (484.0–658.4) | 501.0 (462.8–562.1) | -31.5 (-74.2–38.0) | 0.460 | 0.13 | 542.1 (481.1–612.0) | 482.6 (470.9–556.0) | -46.7 (-81.3–0.62) | **0.002** | 0.52 Large | 0.294 | 0.18 |
| | CRT-CoV* | 0.2 (0.1–0.3) | 0.2 (0.1–0.3) | 0.0 (-0.0–0.10) | 0.842 | 0.04 | 0.2 (0.2–0.3) | 0.2 (0.2–0.2) | 0.0 (-0.1–0.0) | 0.148 | 0.24 | 0.268 | 0.19 |
| | RT*~ | 429.5 (392.3–488.5) | 422.0 (383.3–455.0) | -15.0 (-71.0–23.0) | 0.244 | 0.21 | 444.0 (405.3–504.5) | 428.5 (378.8–497.5) | -23.0 (-38.0–7.0) | 0.069 | 0.30 | 0.820 | 0.04 |
| **Working memory** | FDS | 6.0 (5.0–6.0) | 6.0 (6.0–7.0) | 0.0 (0.0–1.0) | 0.053 | 0.31 Medium | 6.5 (5.3–7.0) | 7.0 (6.0–8.0) | 0.0 (0.0–1.0) | **0.021** | 0.37 Medium | 0.598 | 0.09 |
| | PS~ | 18.5 (14.0–28.8) | 22.0 (8.8–27.8) | 0.0 -11.0–8.0) | 0.900 | 0.02 | 25.5 (14.0–26.0) | 26.5 (15.3–27.8) | 0.0 (-5.0–2.0) | 0.859 | 0.03 | 0.934 | 0.02 |
| **Executive function** | TMT A* | 37.7 (29.7–50.8) | 31.2 (24.0–42.5) | -5.6 (-11.5–0.6) | **0.017** | 0.38 Medium | 39.3 (26.1–54.2) | 36.0 (28.9–50.4) | -1.8 (-3.5–8.3) | 0.520 | 0.10 | **0.040** | 0.33 Medium |
| | TMT B* | 81.2 (61.0–113.7) | 60.5 (47.0–93.9) | -7.2 (-21.8–0.0) | **0.013** | 0.40 Medium | 84.4 (62.0–129.3) | 72.9 (59.8–86.3) | -14.9 (-34.8–3.7) | **0.011** | 0.41 Medium | 0.693 | 0.06 |
| | TMT BA* | 46.9 (22.1–69.9) | 29.6 (23.2–63.4) | -1.2 (-19.8–12.9) | 0.351 | 0.15 | 49.6 (26.1–84.6) | 30.6 (22.7–45.0) | -13.0 (-43.0–0.4) | **0.006** | 0.44 Medium | 0.160 | 0.22 |
| **Visuospatial ability** | JLO | 19.5 (13.3–25.0) | 18.5 (13.5–23.5) | -1.0 (-2.8–1.8) | 0.209 | 0.20 | 20.0 (18.3–23.0) | 22.0 (17.0–24.0) | 1.0 (-2.0–5.0) | 0.347 | 0.15 | 0.161 | 0.22 |
| | EHC*~ | 67233.0 (53361.3–79196.3) | 63403.0 (30497.8–74705.3) | -7132.0 (-1168.0–636.0) | 0.053 | 0.35 Medium | 67807.5 (56020.8–81883.3) | 58288.0 (19865.5–74207.5) | -3251.0 (-17231.0–9775.0) | 0.460 | 0.13 | 0.468 | 0.13 |
| | MOT~ | 0.7 (0.6–0.7) | 0.6 (0.5–0.8) | 0.5 (-0.0–0.1) | 0.148 | 0.26 | 0.7 (0.6–0.8) | 0.7 (0.6–0.8) | 0.0 (-0.1–0.3) | 0.629 | 0.08 | 0.771 | 0.05 |

(*Continued*)

**Table 3.** (Continued)

| Outcome | | TVT group | | | | | Standard group | | | | | Between group difference | |
|---|---|---|---|---|---|---|---|---|---|---|---|---|---|
| | | Pre | Post | Change score | p | Effect size | Pre | Post | Change score | p | Effect size | p | Effect size |
| **Visual outcomes** | | | | | | | | | | | | | |
| **Visual function** | VA* | 0.14 (0.04–0.30) | 0.10 (0.00–0.16) | -0.02 (-0.12–0.02) | 0.073 | 0.30 | 0.12 (0.00–0.20) | 0.06 (0.00–0.14) | -0.04 (-0.10–0.02) | 0.072 | 0.33 Medium | 0.862 | 0.03 |
| | CS | 1.50 (1.33–1.56) | 1.44 (1.36–1.64) | -0.08 (-0.04–0.16) | 0.119 | 0.25 | 1.44 (1.40–1.62) | 1.48 (1.44–1.68) | 0.00 (-0.08–0.04) | 0.952 | 0.01 | 0.087 | 0.30 |
| | VC*~ | 0.17 (0.03–0.28) | 0.11 (-0.07–0.44) | -0.07 (-0.27–0.24) | 0.155 | 0.27 | 0.13 (-0.05–0.40) | 0.17 (0.03–0.28) | 0.00 (-0.08–0.02) | 1.000 | 0.0 | 0.260 | 0.23 |
| | CS-6~ | 1.60 (1.20–1.80) | 1.80 (1.40–1.90) | 0.0 (-0.13–0.45) | 0.573 | 0.11 | 1.90 (1.30–2.10) | 1.80 (1.40–1.90) | -0.10 (-0.38–0.00) | 0.574 | 0.11 | 0.209 | 0.25 |
| | CS-18~ | 0.60 (0.50–0.80) | 0.80 (0.50–0.80) | 0.0 (0.0–0.4) | **0.026** | 0.42 Medium | 0.80 (0.50–1.20) | 0.60 (0.50–1.20) | 0.00 (-0.18–0.00) | 0.750 | 0.06 | **0.033** | 0.42 Medium |
| | NFQ~ | 5.0 (1.5–8.0) | 4.5 (2.0–9.8) | -1.0 (4.0–2.0) | 0.570 | 0.10 | 8.0 (6.0–12.3) | 6.0 (3.25–7.75) | -3.0 (-5.0–2.0) | 0.087 | 0.31 Medium | 0.318 | 0.18 |

UPDRS, Unified Parkinson's Disease Rating Scale–Motor; SRT, simple reaction time; CoV, coefficient of variance; CRT, choice reaction time; FDS, Forward Digit Span; TMT, Trail Making test; JLO, Benton's Judgement Line of Orientation Test; VA, visual acuity (LogMAR); CS, contrast sensitivity (MARS); NFQ (SSS), Near far quickness, Senaptec Sensory Station

Numbers in bold represent values of statistical significance p < 0.05

*Lower scores indicate improvement

~Senaptec Sensory Station outcomes: RT, Reaction Time, average time to hit targets (msec); MOT, Multiple Object Tracking, proportional score; EHC, Eye-Hand Co-ordination, total time to hit 80 targets (msec); VC, Visual Clarity; CS-6, Contrast Sensitivity; NFQ, Near Far Quickness, no of correct responses; PS, Perception Span, total score.

There were no significant improvements in visual functions detected on the traditional chart-based tests (LogMAR and MARS contrast sensitivity). However, CS-18 measured on the Senaptec battery showed a medium effect size for improvement in the TVT group compared to the SC group (effect size:0.42, p = 0.033).

Table 4 shows scores for baseline and follow-up, within group and between group changes as a result of the intervention, for motor, clinical and self-report outcomes. Improvements were

**Table 4. Baseline and follow-up scores (median and interquartile range), within group changes and between groups differences (p) for change over time for clinical, quality of life, and motor performance outcomes.**

| Outcome | | TVT group | | | | | Standard group | | | | | Between group difference | |
|---|---|---|---|---|---|---|---|---|---|---|---|---|---|
| | | Pre | Post | Change score | p | Effect size | Pre | Post | Change score | p | Effect size | p | Effect size |
| **Clinical outcome** | | | | | | | | | | | | | |
| | UPDRS* | 39.0 (20.3–48.3) | 21.0 (16.3–35.0) | -6.0 (-21.8–1.3) | **0.001** | 0.51 Large | 25.5 (19.3–46.8) | 19.0 (13.0–40.0) | -3.0 (-15.0–1.0) | **0.007** | 0.43 Medium | 0.354 | 0.14 |
| **Self-report outcomes** | | | | | | | | | | | | | |
| **Quality of life** | PDQ-39 SI* | 20.2 (12.6–26.7) | 20.5 (9.3–25.3) | -2.5 (-5.5–1.7) | 0.079 | 0.28 | 27.7 (15.7–39.1) | 20.0 (12.4–30.3) | -2.8 (-9.5–1.7) | **0.043** | 0.33 Medium | 0.421 | 0.09 |
| **Falls self-efficacy** | FES-I* | 26.0 (19.3–31.0) | 23.5 (19.3–31.5) | 1.0 (-4.0–6.0) | 0.657 | 0.07 | 30.0 (21.0–44.8) | 24.0 (18.0–40.0) | -2.0 (-9.0–0.0) | **0.046** | 0.32 Medium | 0.137 | 0.28 |
| **Fatigue** | FSS* | 32.0 (23.8–41.5) | 29.0 (17.3–38.0) | -2.5 (-6.5–5.8) | 0.525 | 0.10 | 41.0 (30.0–50.0) | 43.0 (29.0–49.0) | 3.0 (-2.8–7.3) | 0.234 | 0.19 | 0.149 | 0.20 |
| **Motor performance outcomes** | | | | | | | | | | | | | |

(*Continued*)

**Table 4.** (Continued)

| Outcome | | TVT group | | | | | Standard group | | | | | Between group difference | |
|---|---|---|---|---|---|---|---|---|---|---|---|---|---|
| | | **Pre** | **Post** | **Change score** | **p** | **Effect size** | **Pre** | **Post** | **Change score** | **p** | **Effect size** | **p** | **Effect size** |
| **Physical function** | MBT | 20.5 (17.3–26.0) | 24.0 (20.5–26.8) | 1.0 (0.0–4.8) | **0.006** | 0.43 Medium | 16.0 (14.0–24.0) | 21.0 (18.0–27.0) | 2.5 (0.8–4.0) | **0.007** | 0.44 Medium | 0.842 | 0.13 |
| | TUG* | 12.3 (10.3–14.7) | 11.6 (10.3–14.0) | -0.5 (-2.3–0.5) | 0.136 | 0.24 | 14.23 (12.5–17.9) | 13.1 (11.0–17.6) | -0.1 (-3.4–1.1) | 0.356 | 0.15 | 0.993 | 0.01 |
| **Gait (single task)** | Gait Speed (m/s) | 0.91 (0.8–1.16) | 0.95 (0.84–1.12) | 0.01 (-0.05–0,09) | 0.344 | 0.15 | 0.8 (0.6–1.0) | 0.9 (0.7–1.1) | 0.02 (-0.01–0.07) | 0.084 | 0.28 | 0.557 | 0.10 |
| | Foot Strike Angle (deg) | 20.27 (13.66–23.09) | 16.72 (14.57–23.67) | 0.21 (-1.73–2.54) | 0.658 | 0.07 | 13.7 (11.6–20.3) | 15.5 (12.1–22.9) | 0.18 (-1.05–3.04) | 0.356 | 0.15 | 0.537 | 0.10 |
| | Stride Length (m) | 2.15 (1.84–2.42) | 2.06 (1.89–2.41) | 0.06 (-0.03–0.12) | 0.184 | 0.21 | 1.9 (1.6–2.3) | 2.0 (1.7–2.3) | 0.03 (-0.01–0.12) | 0.068 | 0.30 | 1.000 | 0.00 |
| | Stride Length SD (m) | 0.09 (0.07–0.10) | 0.09 (0.73–0.12) | 0.01 (-0.01–0.02) | 0.166 | 0.22 | 0.1 (0.1–0.1) | 0.1 (0.1–0.1) | 0.00 (-0.02–0.02) | 0.709 | 0.06 | 0.25 | 0.19 |
| | Stride Time (s) | 1.14 (1.03–1.27) | 1.13 (1.04–1.22) | -0.01 (-0.05–0.02) | 0.338 | 0.15 | 1.2 (1.1–1.2) | 1.2 (1.1–1.3) | -0.02 (-0.04–0.03) | 0.463 | 0.12 | 0.837 | 0.03 |
| | Stride Time SD (s) | 0.03 (0.02–0.04) | 0.03 (0.02–0.04) | 0.00 (-0.01–0.02) | 0.551 | 0.10 | 0.0 (0.0–0.1) | 0.0 (0.0–0.1) | 0.00 (-0.01–0.01) | 0.634 | 0.08 | 0.821 | 0.04 |
| | Turn Velocity (deg/s) | 147.35 (120.51–167.24) | 153.39 (118.91–171.42) | 2.87 (-9.93–15.45) | 0.376 | 0.14 | 140.9 (113.1–163.5) | 128.0 (112.8–155.9) | -3.35 (-11.22–8.62) | 0.831 | 0.04 | 0.496 | 0.11 |
| **Balance (eyes open, firm)** | Sway Centroidal Frequency (Hz) | 0.95 (0.78–1.21) | 0.88 (0.76–1.05) | -0.07 (-0.19–0.10) | 0.398 | 0.14 | 0.82 (0.67–0.93) | 0.83 (0.68–1.06) | 0.03 (-0.06–0.25) | 0.287 | 0.18 | 0.116 | 0.26 |
| | ML Sway Frequency (Hz) | 1.32 (0.97–1.57) | 1.37 (1.00–1.68) | -0.05 (-0.23–0.02) | 0.494 | 0.23 | 1.26 (0.81–1.42) | 1.22 (0.92–1.57) | 0.05 (-0.04–0.14) | 0.356 | 0.15 | 0.377 | 0.15 |
| | AP Sway Frequency (Hz) | 0.85 (0.70–1.01) | 0.75 (0.66–0.94) | -0.06 (-0.33–0.18) | 0.145 | 0.11 | 0.70 (0.61–0.81) | 0.75 (0.59–0.90) | 0.07 (-0.12–0.24) | 0.163 | 0.23 | **0.025** | 0.37 Medium |
| | ML Jerk | 2.20 (1.06–5.33) | 2.26 (1.10–6.07) | -1.54 (-4.54–0.79) | 0.469 | 0.12 | 1.60 (1.08–5.49) | 2.46 (1.22–6.48) | -0.41 (3.52–3.95) | 0.177 | 0.22 | 0.113 | 0.26 |
| | AP Jerk | 7.41 (2.14–12.60) | 5.38 (2.47–10.00) | -0.17 (-2.65–0.32) | 0.171 | 0.22 | 7.60 (2.25–12.37) | 3.71 (2.46–13.08) | 0.36 (-0.32–1.57) | 0.943 | 0.01 | 0.329 | 0.16 |
| | ML RMS (m/s^2) | 0.03 (0.02–0.04) | 0.03 (0.02–0.05) | -0.01 (-0.03–0.02) | 0.365 | 0.32 Medium | 0.03 (0.02–0.05) | 0.03 (0.02–0.06) | 0.00 (-0.03–0.03) | 0.687 | 0.15 | 0.428 | 0.13 |
| | AP RMS (m/s^2) | 0.09 (0.06–0.14) | 0.07 (0.07–0.12) | -0.00 (-0.02–0.00) | 0.629 | 0.16 | 0.12 (0.07–0.13) | 0.10 (0.07–0.11) | 0.00 (-0.01–0.03) | 0.740 | 0.27 | 0.692 | 0.07 |
| | ML Velocity (m/s) | 0.27 (0.15–0.32) | 0.15 (0.12–0.39) | -0.07 (-0.41–0.18) | **0.049** | 0.32 Medium | 0.20 (0.14–0.31) | 0.22 (0.12–0.36) | 0.02 (-0.06–0.34) | 0.356 | 0.07 | **0.041** | 0.34 Medium |
| | AP Velocity (m/s) | 0.71 (0.02–0.04) | 0.55 (0.34–1.17) | -0.05 (-0.18–0.01) | 0.314 | 0.08 | 0.66 (0.41–0.94) | 0.73 (0.49–1.01) | 0.06 (-0.10–0.15) | 0.102 | 0.06 | 0.113 | 0.26 |

UPDRS, Unified Parkinson's Disease Rating Scale–Motor; MBT, Mini Best Test; TUG, Timed up and Go test (secs); PDQ-39 SI, Parkinson's disease Questionnaire– 39 Single index score; FES-I, Falls Efficacy Scale-I; FSS, Fatigue Severity Scale; MBT, Mini Best Test; TUG, Timed up and Go test (secs).

Numbers in bold represent values of statistical significance p < 0.05

*Lower scores indicate improvement

Numbers in bold represent values of statistical significance p < 0.05

observed in motor performance outcomes across both groups. The TVT group showed large effects for improvements in MDS UPDRS-motor score (effect size: 0.51; p = 0.001) and medium effects for improvement in the Mini-BESTest (effect size: 0.43; p = 0.006). The SC group showed similar improvements with a medium effect in the UPDRS-motor score (effect size: 0.43; p = 0.007) and in the Mini-BESTest (effect size: 0.44; p = 0.007). Furthermore, the TVT group showed medium effects for improvements in mediolateral (ML) sway velocity (effect size: 0.32; p = 0.049) and ML root mean square (RMS) value (effect size: 0.32; p = 0.365).

The SC group showed medium effects for improvements in QoL (PDQ-39 score effect size: 0.33; p = 0.043) and falls efficacy (FES-I score effect size: 0.32; p = 0.046).

## Discussion

This pilot study evaluated the feasibility of multimodal TVT in people with PD compared to SC. The novelty of our study is the combined use of stroboscopic visual training with a digital rehabilitation tool (mobile application) to deliver a home-based visuo-cognitive intervention for people with PD. The main findings suggested that 8 sessions of supervised, home-based TVT were acceptable and feasible to deliver in people with PD. Preliminary efficacy findings suggest that TVT may improve visual, cognitive and motor outcomes in PD similar to SC, but a larger sample is needed to confirm results and examine whether TVT is superior to SC.

### Feasibility of study design

**Recruitment.** The present study showed a good recruitment rate which was expected given that both arms of the study involved an active (not placebo) intervention [67]. Almost two thirds of all participants were recruited directly from movement disorder clinics where members of the research team were in attendance in their capacity as honorary physiotherapists. The collaboration between hospital-based clinicians and researchers facilitated recruitment in this study. This has been identified as a key factor for successful enrolment in previous literature [68]. Despite a rigorous screening protocol, two participants were identified as having a history of light-sensitivity which was not reported until the end of the baseline assessment. These participants remained in the study but were allocated to the SC arm to avoid the potential risks of exercising in stroboscopic conditions. Given this, a more rigorous screening of light sensitivity is recommended for any future studies involving stroboscopic glasses, to ensure true randomisation.

Retention rate and adherence to both arms of the intervention were high, with rates comparable with those of other home-based exercise interventions [69]. There was no difference in retention and adherence rates between groups. This finding further supports the benefits of having an active comparator rather than a placebo group and is contextualised by data from the interviews which showed that both interventions were well received by participants [70]. The only participant to drop out did so immediately after the baseline testing (prior to being informed of group allocation), due to personal issues. The timing of recruitment (from July 2021 to November 2022) coincided with the ongoing COVID-19 pandemic which resulted in a transition to remote service delivery and disrupted community care [71,72]. Therefore, an additional factor contributing to successful retention in this study may have been the face-to-face access to a physiotherapist, at home, for support and motivation during the study.

### Feasibility of TVT

This trial was novel in its combined use of stroboscopic glasses and a cognitive training application to form a multimodal TVT intervention. To our knowledge, the only previous study to have utilised stroboscopic glasses in a neurological cohort involved a single training session in

people with multiple sclerosis and did not report on any safety or usability issues [44]. In light of the paucity of evidence around the feasibility of stroboscopic glasses as a training tool in this population, we established the need for direct supervision from a physiotherapist during every visit. Not only did this contribute to excellent retention and adherence rates as previously discussed, it also allowed interventions to be tailored according to each individual, provision of technological support as required, and monitoring of the effects of the stroboscopic training.

Seven participants experienced symptoms associated with use of the stroboscopic glasses which were largely triggered by activities involving head turns or whole-body rotational exercises. These symptoms resolved after short rest periods and could be avoided by adapting the exercises. As this is the first trial to report symptoms associated with the use of stroboscopic glasses in any population, there are no figures to compare with other trials. However, the nature of the symptoms experienced by participants are in keeping with the symptoms associated with real-induced motion sickness [73] or those reported during immersion in virtual reality environments [74]. These findings indicate that supervision is required when undertaking stroboscopic visual training in PD and exercises may need to be tailored to avoid inducing symptoms. One fall did occur during the study period in the TVT group, but this was mechanical in nature and occurred outside of intervention treatment sessions. As people with PD are more likely to fall due to the prevalence of postural instability, this outcome is not unexpected [75]. Reassuringly, no falls occurred during the intervention visits, which further supports the benefits of one-to-one supervision when undertaking TVT.

## Preliminary efficacy outcomes

Although the primary focus of this study was on feasibility of TVT, we performed preliminary analysis to compare TVT and SC. While these findings should be considered with caution due to the small sample sizes, both interventions positively influenced a variety of cognitive, visual, and motor performance outcomes in people with PD, emphasising the therapeutic value of the SC intervention as well as the TVT.

Within the cognitive domains, the TMT were the most sensitive indicators for improvements in rote memory and executive function, with medium effect sizes across both study arms. Improvements in working memory were also detected on the FDS in both groups. These data are in keeping with the findings of a meta-analysis on the use of cognitive training in PD which showed significant improvements in specific cognitive domains including working memory and executive functions [76]. Computer-based CRT improved in both groups, although with a much greater effect size in the SC group. This suggests that cognitive improvements were not specific to the mode of training received.

No significant improvements were detected in the standard chart-based tests of visual function as a result of the training. It is worth noting that improvements in CS were detected in the TVT group on the SSS. While this could suggest that the computerised battery was more sensitive to change than the standardised MARS CS chart, this result should be interpreted with caution due to the current lack of validation of the SSS in PD.

Health-related QoL as measured with the PDQ39 showed positive trends in the SC group compared to the TVT group. The nature of the training may have contributed to this finding, as participants were actively engaging with the researcher during the paper-based activities and games, whereas those in the TVT group were working on an iPad with less interaction [77]. This is contextualised by data from the interviews which showed the value that people with PD placed on the therapy interaction that accompanied the training sessions (qualitative findings published elsewhere) [70]. Future work needs to determine how much influence social engagement has on the uptake and integration of technology into rehabilitative practices

[70]. Fear of falling as measured by the FES-I also showed a greater degree of improvement in the SC group. Use of the stroboscopic glasses in the TVT group may have resulted in an increased fear of falling due to the disruption to vision during exercises which may have contributed to the result.

Within the motor domains, the MDS UPDRS-III and Mini-BESTest detected improvements with medium to large effect sizes in both groups. Between-group differences with medium effect sizes were seen in several of the sway metrics (e.g., change score for anteroposterior (AP) sway frequency and ML sway velocity) which suggest that the TVT intervention had a greater impact on balance than the SC intervention.

It has long been recognised that increased postural sway is a risk factor for falling in people with PD [78–81] and previous work has shown there is an association between static measures of balance and visuo-cognition (attention and visuospatial functions) [6,14]. The added value of the technology intervention might be explained by the nature of the training. The TVT required participants to exercise in suboptimal visual conditions which implicitly trained the somatosensory system and demanded more focus and attention than that which was required of participants exercising in the SC group.

Results from several studies have shown that interventions that enhance visuo-cognitive skills lead to improvements in fall risk factors [82–85]. Our findings are consistent with these observations and further support the notion that a multimodal visuo-cognitive intervention could be beneficial to reduce fall rates in people with PD.

Although modest changes were seen in both groups, it could be argued that these changes are still clinically significant given that PD is a progressive disease for which any benefits to mobility and QoL can be considered to be helpful [86]. In keeping with previous studies [77,87], our results suggest that visuo-cognitive training benefits in people with PD differ depending on the mode of intervention. In this respect, multimodal visuo-cognitive training using a combination of technology and standard care approaches should be considered depending on the needs and preferences of the individual with PD.

## Sample size estimations for future trails

Mean change scores and standard deviations of the TMT A, UPDRS and CS-18 were used to calculate a sample size for a future large trial. We aimed to detect an effect size of 0.8, allowing for 20% dropouts, based on the assumption of two groups of equal size and a two tailed significance threshold alpha of 0.05. An estimated recruitment target of 494 participants (247 in each group) would be required to demonstrate clinical efficacy in a future trial. While this figure is in keeping with previous large scale RCTs [88,89], consideration must be given to whether the delivery of home-based TVT is feasible for a sample of this size. Future studies would need to consider the cost implications of factoring in multisite data collection to ensure availability of staff and adequate equipment to deliver the face-to-face intervention.

Based on the variables which showed the largest effect sizes in this study, we recommend prioritising the following outcomes in future trials of TVT: computerised assessment of reaction time and eye-hand co-ordination, FDS, TMT, UPDRS-III, Mini-BESTest and specific balance metrics (AP sway frequency, ML RMS and ML velocity) from wearables. Inclusion of QoL measures and fear of falling should also be considered in future studies, given the difference in findings between the two intervention approaches.

## Strengths and Limitations

The greatest strength of this study is the combined use of stroboscopic glasses with a mobile application to provide comprehensive visuo-cognitive rehabilitation. This is more

representative of what would be carried out in multidisciplinary clinical practice [90]. The rigorous methodology explored both the feasibility of the study design and technological intervention, while also including a multitude of paper-based and computerised outcome assessments with comparisons to standard care. Using an active comparator group enabled a more realistic exploration of recruitment and retention rates, as well as randomization procedures [91]. Furthermore, the TVT and SC group interventions were designed to be equivalent on as many elements as possible, thereby enabling factors such as social support to be controlled for [92]. A further strength is in the design of the study, which involved twice weekly visits over four weeks. A previous review of physiotherapy interventions revealed that more intensive (i.e. >3 times per week) and long duration interventions may not be feasible in clinical practice [93]. Our trial was designed such that TVT would be transferable to mainstream healthcare (i.e., a feasible number of interventions and time period for application in clinical practice), although the ideal number and duration of sessions requires additional research. Due to the novel, exploratory nature of this study, a therapist was in attendance at every session. Moving forwards, it will be necessary to consider how much therapist involvement is needed to deliver an exercise programme using novel technology rather than SC. This is particularly relevant if the ultimate goal is to facilitate self- management [94]. Future studies using TVT should therefore show the costs and outcomes adjusted for differential resource needs in the context of rehabilitation for people with PD [95,96].

This study is novel in its use of stroboscopic technology as part of a multimodal rehabilitation intervention for people with PD. Stroboscopic training has never previously been used in this population, despite evidence for its potential benefits in clinical practice [97]. However, this also presented some methodological limitations. We acknowledge that flaws in the screening process resulted in two participants being placed in the SC group as a precautionary measure after revealing they had experienced some light sensitivity in the past. Given the lack of existing research, it is not possible to determine whether these individuals would have been affected by the stroboscopic effect, but the decision to move them to the SC group in this study represents a lack of true randomisation. For future studies involving stroboscopic training in PD, we suggest the screening process could be strengthened to avoid such issues by adding objective evaluations of photosensitivity and motion sickness susceptibility [98,99]. Furthermore, it is not known whether accommodation to stroboscopic conditions over time could lead to a reduction in symptoms such as nausea and dizziness [100]. In the absence of existing stroboscopic training research, the threshold for discontinuing use of the stroboscopic glasses in response to the onset of symptoms was based on clinical judgement. There were occasions when participants were willing to continue the intervention despite experiencing symptoms, but were advised to rest and remove the glasses by the treating physiotherapist as a precautionary measure. It is therefore possible that the stroboscopic training was stopped prematurely in some participants who may have been able to tolerate the symptoms and progress further with the exercises, thereby impacting on results. Use of an objective measure of symptoms (e.g. Simulator Sickness Questionnaire [101]) would allow the detection of changes in symptoms of nausea, dizziness or disorientation due to exposure to the stroboscopic conditions [102].

Given the potential to experience negative symptoms as a result of the stroboscopic effect, it is essential that future trials are conducted to establish the clinical efficacy of stroboscopic training in PD (and whether it is superior to exercise under normal visual conditions as part of a TVT package) in order to facilitate future user-adoption. Negative symptoms may prove to be an exclusion criteria for use of the strobes in clinical practice, or may indicate the need to use lower strobe settings to build up tolerance. Future work should explore the impact of stroboscopic training length on symptom severity over time, and establish guidelines for their use in clinical practice.

There were also a number of other limitations. First, this was a pilot study that focussed primarily on the feasibility of TVT in relation to important practicalities such as recruitment, participant retention and treatment fidelity. It was not adequately powered for a specific primary outcome, and did not include correction for multiple testing. While we have provided efficacy results, these are only preliminary findings and should therefore be treated with caution due to the potential problems arising from the small sample sizes and low statistical power [103]. Second, blinding of participants and the researcher delivering the training was not possible due to the nature of the interventions, and therefore the trial was liable to potential performance bias [93]. A more rigorous blinding process would be necessary for a definitive larger-scale trial. Third, as our outcome measures were assessed shortly after the intervention had ceased, we cannot provide insight into the long-term effect of TVT. Fourth, in the absence of a "gold standard" approach to visuo-cognitive training, a non-technological approach was designed to contribute scientific legitimacy to the methodology by providing a comparative arm to the study. However, while this was labelled as "standard care" because it contained a variety of non-technological interventions that have been used in previous studies and within clinical practice, the reality is that there is no "gold standard" package of visuo-cognitive training interventions. The findings from this study invite ensuing studies to improve upon the design and utilise new technology and visual training tools to further investigate this emerging area of multimodal rehabilitation in PD.

## Conclusion

The current study responds to the need for new and improved interventions to address visuo-cognitive dysfunction and contributes to the evidence base for multimodal approaches to rehabilitation in PD. Furthermore, this study not only provides a novel insight into the use of strobe glasses as an adjunct to exercise in PD, it is also the first time that a comprehensive technological visuo-cognitive training programme has been used in this population. Our study demonstrates that home-based TVT is feasible with therapy support and the components of the intervention warrant further exploration.

## Supporting information

**S1 File. CONSORT checklist.**
(DOCX)

**S2 File. Study protocol.**
(PDF)

**S3 File. Outline of interventions.**
(DOCX)

**S4 File. Description of the visuo-cognitive training interventions using TIDieR framework.**
(DOCX)

## Acknowledgments

We would like to thank the participants for generously giving up their time to take part in this study.

## Author Contributions

**Conceptualization:** Julia Das, Gill Barry, Rosie Morris, Samuel Stuart.

**Data curation:** Julia Das, Gill Barry.

**Formal analysis:** Julia Das, Gill Barry, Richard Walker, Rodrigo Vitorio, Paul Oman, Samuel Stuart.

**Funding acquisition:** Samuel Stuart.

**Investigation:** Julia Das, Rodrigo Vitorio, Yunus Celik.

**Methodology:** Julia Das, Gill Barry, Richard Walker.

**Project administration:** Julia Das.

**Resources:** Richard Walker, Claire McDonald, Bryony Storey.

**Supervision:** Gill Barry, Richard Walker, Rosie Morris, Samuel Stuart.

**Validation:** Julia Das, Gill Barry.

**Visualization:** Julia Das, Gill Barry.

**Writing – original draft:** Julia Das.

**Writing – review & editing:** Julia Das, Gill Barry, Richard Walker, Rodrigo Vitorio, Yunus Celik, Claire McDonald, Bryony Storey, Paul Oman, Rosie Morris, Samuel Stuart.

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
