## [Decision Letter · Decision Letter 0]

11 Jul 2024

PDIG-D-24-00019

The feasibility of a visuo-cognitive training intervention using a mobile application and exercise with stroboscopic glasses in Parkinson’s: Findings from a pilot randomised controlled trial

PLOS Digital Health

Dear Dr. Das,

Thank you for submitting your manuscript to PLOS Digital Health. After careful consideration, we feel that it has merit but does not fully meet PLOS Digital Health's publication criteria as it currently stands. Therefore, we invite you to submit a revised version of the manuscript that addresses the points raised during the review process.

Please submit your revised manuscript within 60 days Sep 09 2024 11:59PM. If you will need more time than this to complete your revisions, please reply to this message or contact the journal office at digitalhealth@plos.org. Please include the following items when submitting your revised manuscript:

We look forward to receiving your revised manuscript.

Kind regards,

Walter Karlen

Academic Editor

PLOS Digital Health

Journal Requirements:

Additional Editor Comments (if provided):

Dear authors

thank you or your submission to PLOS digital health. Please find attached the reviewers comments. A major question is whether your currently presented work is substantially different than the already previously published work (in this and other journals). Please clarify in your manuscript what the new contributions are and provide more details in methods, results and discussion to support those claims. Please also add a rationale, supported by existing literature, that would have lead to your feasibility study. 

On a last note, there is the open questions why the authors did not try to submit these study results in the same journal where they published the protocol?

Reviewers' comments:

Reviewer's Responses to Questions

**Comments to the Author**

1. Does this manuscript meet PLOS Digital Health’s publication criteria? Is the manuscript technically sound, and do the data support the conclusions? The manuscript must describe methodologically and ethically rigorous research with conclusions that are appropriately drawn based on the data presented.

Reviewer #1: Partly

Reviewer #2: Yes

2. Has the statistical analysis been performed appropriately and rigorously?

Reviewer #1: Yes

Reviewer #2: Yes

3. Have the authors made all data underlying the findings in their manuscript fully available (please refer to the Data Availability Statement at the start of the manuscript PDF file)?

Reviewer #1: Yes

Reviewer #2: Yes

4. Is the manuscript presented in an intelligible fashion and written in standard English?

Reviewer #1: Yes

Reviewer #2: Yes

5. Review Comments to the Author

Reviewer #1: The current manuscript seems to be the second installment of a previously published research proposal. Therefore reviewing this manuscript required reading through the previous published paper in detail as well, and it would seem that readers will need to connect the two in order to have a full understanding of the project. While overall the research question and goals are interesting and the methodology is presented well with satisfactory detail, there is a lack of connection on the importance or rationale for using the technology based program. Some specific questions are below.

1. What is the benefit of the mobile device/technology over and above the standard pen and paper or 'real world' tasks? 

2. Why is the use of stroboscopic glasses considered a particular strength in the study? This was inadequately addressed in the previous publication as well. What would this kind of training add over and above what may already be done in standard care "SC"? Where would you expect the impact to be (e.g. what variables or measures would this have particular influence on)? Did you see any evidence of this? Other than stating it is a strength, in reading the manuscript I do not see where that strength may be or where the results support this statement.

3. I realize this is a feasibility study and it seems that the result of the study is yes, technology can be added in a feasible way to at home training. But again why is this desired? What benefit does this have over and above SC? Based on some of the QOL data of this study it seems that potentially the SC has more benefits if we consider QOL and potential social engagement of the PwP in their training. This result is downplayed a little with more emphasis on the benefits of the technology.

4. In addition to the above, the stroboscopic goggles may also have contributed to some motion sickness related discomfort. Again, this brings me to question why use these if there is no further benefit to SC training but brings with it added risk?

Reviewer #2: This paper presents preliminary work toward a larger goal of understanding risks and benefits of a novel visual-cognitive training intervention to improve balance and function in individuals with Parkinson's disease (PD). It's strength is that the authors do not attempt to overestimate their findings, showing the feasibility of the intervention but understanding their preliminary results to not show improvement over standard care. The manuscript is well-written, with appropriate detail in both methodology and results. There are no issues noted in the writing and figures are clear and add to the paper. The only minor issue is that there could be more discussion about cost-benefit analysis and the role of the therapist for each of the treatments. It appears that the TVT intervention may require greater therapist involvement (given some increase in side-effects and need for treatment modifications in this group) which would raise the cost of providing this treatment compared to usual care. This is particularly relevant if the goal is an in-home training program. Other than that, the paper adds new information to the field of visual-motor training in PD and addresses an important, unmet need in this population.

6. PLOS authors have the option to publish the peer review history of their article (what does this mean?). If published, this will include your full peer review and any attached files.

**Do you want your identity to be public for this peer review?** For information about this choice, including consent withdrawal, please see our Privacy Policy.

Reviewer #1: No

Reviewer #2: Yes: Lisa M. Muratori

---

## [Editor Report · Decision Letter 1]

10 Nov 2024

The feasibility of a visuo-cognitive training intervention using a mobile application and exercise with stroboscopic glasses in Parkinson’s: Findings from a pilot randomised controlled trial

PDIG-D-24-00019R1

Dear Dr Das,

We are pleased to inform you that your manuscript 'The feasibility of a visuo-cognitive training intervention using a mobile application and exercise with stroboscopic glasses in Parkinson’s: Findings from a pilot randomised controlled trial' has been provisionally accepted for publication in PLOS Digital Health.

Best regards,

Walter Karlen

Academic Editor

PLOS Digital Health

The authors have addressed all reviewer comments. They added a large number of literature to support their approach. Please remove "rigorous" from l 548 as this is not quantifiable.